# Optimizing Treatment Precision: Role of Adaptive Radiotherapy in Modern Anal Cancer Management

**DOI:** 10.3390/cancers17152478

**Published:** 2025-07-26

**Authors:** David P. Horowitz, Yi-Fang Wang, Albert Lee, Lisa A. Kachnic

**Affiliations:** 1Department of Radiation Oncology, Columbia University Irving Medical Center, New York, NY 10032, USA; 2Herbert Irving Comprehensive Cancer Center, New York, NY 10032, USA

**Keywords:** anal cancer, radiotherapy, adaptive radiation

## Abstract

Anal cancer that is not metastatic is treated with definitive chemoradiation. The introduction of intensity-modulated radiation therapy significantly reduced treatment toxicity; adaptive radiotherapy is a strategy that may further improve treatment precision and individualize radiation treatment for patients. This review discussed the role of adaptive radiotherapy in the treatment of anal cancer, beginning with an overview of the principles of adaptive radiation, describing the technical workflows, and reviewing the clinical evidence for its use in treatment of anal cancer.

## 1. Introduction

Anal cancer is a relatively rare malignancy, but its incidence has been rising over the past two decades, with 10,930 new cases diagnosed in 2025 in the United States [1]. In contrast to most other gastrointestinal tumors, anal squamous cell carcinoma (SCC) has a predilection for locoregional spread with a lower propensity for distant metastasis [2]. Consequently, definitive chemoradiotherapy (CRT) is the standard curative treatment for localized anal SCC, using concurrent 5-fluorouracil (or capecitabine) and mitomycin alongside radiation [2,3]. This organ-preserving approach yields high cure rates and obviates the need for abdominoperineal resection in most patients, but it can incur substantial acute toxicity (dermatitis, mucositis, diarrhea) and late complications (anal stenosis, incontinence, sexual dysfunction) [4]. Impaired anorectal function is impacted by radiation dose and can significantly impact quality of life for patients treated for anal cancer [5,6,7]. Modern intensity-modulated radiation therapy (IMRT) with daily image guidance has become the standard of care in order to reduce normal tissue dose, yet treatment-related side effects remain a major challenge [2,8,9]. Additional methods for focal dose escalation, such as brachytherapy boost, have shown promising effectiveness but are available at limited centers [10]. Radiation-associated gastrointestinal and dermatologic toxicity may necessitate treatment breaks, which in turn may compromise disease control if unmitigated [2].

Adaptive radiotherapy (ART) has emerged as a strategy to further enhance treatment precision and individualize therapy in response to patient-specific changes during the course of CRT [11]. The rationale for ART in anal cancer stems from the recognition that significant anatomic and tumor changes can occur throughout the 5–6 week treatment course, including tumor shrinkage, weight loss, and variable rectal/bladder filling. In conventional radiation planning, generous margins are added around the tumor to account for such changes, but this comes at the cost of larger irradiated volumes and increased normal tissue dose. Indeed, substantial organ motion in the pelvis (due to bowel gas, bladder filling, etc.) necessitates a wide planning target volume (PTV) margin to ensure adequate coverage, which directly contributes to toxicity through irradiation of small bowel and anterior pelvic contents [12]. By adapting the radiation plan to the patient’s current anatomy, ART offers the potential to tighten margins and thereby spare healthy tissues without sacrificing tumor coverage [13,14]. This is especially pertinent in anal cancer, where reducing the dose to organs-at-risk (OARs) like the small bowel, bladder, skin, and genitalia could lower acute gastrointestinal and dermatologic side effects. Ultimately, adaptive radiotherapy aims to maximize the therapeutic ratio—delivering the right dose to the tumor at each fraction while minimizing collateral damage—which aligns with ongoing efforts to preserve quality of life in these patients.

This review discusses the role of ART in contemporary anal cancer management. We overview the principles of ART, delineate the technical workflows (including both computed tomography (CT) and MR-guided approaches), and examine how adaptive techniques are applied in treatment planning and delivery. We also review the clinical evidence to date, including dosimetric studies and emerging clinical trial data on ART in anal cancer, particularly its impact on outcomes and toxicity. Through this review, we aim to provide a comprehensive understanding of how adaptive radiotherapy is poised to optimize treatment precision for anal cancer and improve patient outcomes.

## 2. Fundamentals of Adaptive Radiotherapy

Adaptive radiotherapy is broadly defined as a feedback process that uses imaging acquired during the treatment course to re-optimize the radiation plan in response to anatomic changes, with the goals of improving target coverage and reducing toxicity [11]. First conceptualized in the late 1990s, ART represents a paradigm shift from the static treatment plan defined at a single time point before a patient begins treatment to a dynamic approach tailored to the patient’s evolving anatomy. In standard non-adaptive radiotherapy, the treatment plan is based on a single pre-treatment CT simulation scan and is applied unchanged for all fractions, relying on uniform PTV margins to account for uncertainties. ART, by contrast, acknowledges that significant inter-fraction and intra-fraction anatomic changes—such as tumor regression, organ motion, or patient weight loss—can occur and negatively impact the original plan, and that these changes can be accounted for through the use of rapid replanning. By adjusting the plan to the current anatomy, ART aims to maintain dosimetric fidelity to the targets while sparing normal tissues beyond what is achievable with initial margins alone [11,13]. ART’s core principle is personalization and precision over time; the treatment is not only personalized to the individual patient, but also to the patient’s day-to-day anatomic variation.

Three general approaches to ART exist: offline, online, and real-time adaptation [11]. Offline ART refers to adaptations made between treatment sessions. This could entail plan recalculation or re-optimization after a few fractions or at a mid-point in the treatment course, with the new plan applied in subsequent sessions. Offline adaptation has been the most traditionally explored method, as it can be implemented within the conventional workflow (e.g., performing a re-simulation and replan while continuing treatment with the original plan until replanning is complete). One offline strategy used in pelvic tumors is the “plan-of-the-day” or library approach, in which multiple plans are created upfront (for example, plans corresponding to small, medium, and large bladder volumes) and the most appropriate plan is selected each day based on the patient’s anatomy [15,16]. Offline ART is well-suited for accounting for relatively slow, systematic changes such as progressive tumor shrinkage or weight loss over weeks. It allows for selective adaptation with minimal disruption—for instance, a single mid-treatment replan if the tumor regresses significantly—and does not require extensive new infrastructure. However, offline ART cannot readily address random daily variations in anatomy (e.g., day-to-day rectal filling differences), without the utilization of a relatively large number of pre-generated “plans of the day” and defining optimal timing or thresholds for offline replanning remains an area of study [11].

Online ART (OART), in contrast, denotes plan adaptation performed in real-time while the patient is on the treatment table, typically immediately prior to delivering each fraction [11]. In an online workflow, imaging is obtained just before treatment each day, the current anatomy is evaluated against the reference plan, and if discrepancies are significant, the plan is adjusted (re-contoured and re-optimized) on the spot. The patient then receives that adapted plan for that fraction. OART directly tackles the challenge of unpredictable, day-to-day anatomic changes, ensuring that each fraction’s dose distribution conforms to that day’s anatomy. The obvious advantage of online adaptation is maximal precision—effectively, adaptive correction at every fraction—which can account for random shifts in organ filling or interfraction tumor changes that offline methods would miss. The trade-off is that online ART places substantial time pressure and resource demands on the clinical workflow, as all steps (imaging, contour propagation/editing, dose calculation, quality assurance) must be completed within minutes, while the patient remains immobilized on the treatment table [11]. Recent technology advancements, including faster imaging and AI-driven automation, have made online ART feasible in routine practice, as discussed later in this review.

Lastly, real-time ART involves continuous adaptation during radiation delivery, such as gating or beam tracking to compensate for intrafraction motion like respiration [17]. Real-time ART includes elements of OART and intrafraction image guidance techniques that are used for non-adaptive image guidance. Table 1 summarizes the key differences between offline, online, and real-time ART approaches.

## 3. Radiation Delivery Techniques

Although the concept of ART has been proposed for decades, its clinical implementation has been limited by technical constraints, particularly for online and real-time ART. These approaches require that imaging, contour delineation, and treatment planning all be performed while the patient is positioned on the treatment couch. This necessitates the integration of advanced online imaging modalities and rapid replanning systems with linear accelerator delivery platforms. Today, two major techniques have emerged based on their imaging guidance methods: MRI-guided and CBCT-guided adaptive radiotherapy systems.

MRI-guided ART initially began with offline workflows, which involved MRI simulation using flat tabletops, laser positioning systems, and MR-compatible immobilization devices. Two major simulation workflows have been implemented: the MR-only workflow and the combined CT-MR simulation approach. In the MR-only workflow, synthetic CT (sCT) images are generated from single or multi-sequence MRI datasets, providing both high geometric accuracy and CT-equivalent Hounsfield unit (HU) information for dose calculation [18]. The combined approach requires deformable or rigid registration between MRI and planning CT images; however, it introduces potential issues such as misregistration, anatomical discrepancies between scans, and increased operational costs [19]. The development of MR-guided linear accelerators (MR-linacs) has enabled online adaptive radiotherapy, integrating high-contrast soft tissue imaging with radiation delivery, along with capabilities for real-time motion tracking and gating [20]. Three main commercial systems exist, each using different magnetic field strengths and configurations. The 0.35T system uses a split-solenoid MRI and a 6 MV linac with beam delivery through a vertical magnet gap, supporting real-time imaging and advanced adaptive workflows. The 0.5T system, with a rotating open bi-planar design, allows flexible beam orientation and full 3D tumor positioning, minimizing magnetic field-induced dose perturbations. Lastly, the 1.5T system integrates a 7 MV standing wave linac on a rotating ring gantry with a closed-bore MRI, offering high-precision targeting, real-time motion tracking, and daily plan adaptation. Each platform reflects unique engineering solutions to integrate MR imaging and radiation delivery while managing magnetic field interference and maximizing clinical flexibility [21].

CBCT has long been used for patient positioning and target alignment immediately before beam delivery. While anatomical changes observed on CBCT can prompt offline adaptive replanning, conventional CBCT lacks the image quality of fan-beam simulation CT and is therefore unsuitable for treatment planning. As a result, traditional offline adaptive workflows require patient re-simulation with a simulation CT scanner, delaying adaptive treatments to subsequent fractions. The recent introduction of the Ethos platform (Varian, Palo Alto), a ring-shaped linear accelerator delivering 6 MV flattening filter-free beams, allows advanced CBCT-guided OART. Ethos integrates an enhanced imaging panel, HyperSight CBCT, which supports on-couch contouring, planning, and dose calculation. HyperSight uses large CsI-based detectors, twice the size of conventional panels, offering improved readout speed, reduced lag, and increased x-ray efficiency. A full-fan, half-arc scan can be completed in 6 s. Advanced reconstruction algorithms, including iterative CBCT, Acuros-based and Monte Carlo scatter corrections, and metal artifact reduction (MAR), further improve image quality, even in patients with metal implants. Additionally, the platform offers automated contouring and a template-driven Intelligent Optimization Engine (IOE) for replanning, enabling adaptive plans to be generated within minutes [22,23,24,25].

Online adaptive workflows in both MRI- and CBCT-guided systems follow several key stages: initial CT or MRI simulation, initial contouring and treatment planning, and daily adaptive treatment sessions. Each adaptive session involves daily imaging, recontouring of the target and OARs, treatment replanning, plan review or selection, and treatment delivery. During each adaptive session, MRI-guided systems offer two strategies: Adapt-to-Position (ATP) and Adapt-to-Shape (ATS). ATP plans are generated via rigid registration between the planning image and the daily MRI, whereas ATS plans involve recontouring the updated target anatomy and generating a new treatment plan based on that shape. A similar approach is employed in the Ethos CBCT-guided system, where the user selects either a scheduled plan (rigidly registered to daily CBCT) or an adaptive plan (re-optimized for the daily anatomy). Beyond its superior soft tissue contrast and potential for biologic imaging, a key advantage of MRI-guided radiotherapy is the ability to perform real-time imaging and gating. In this approach, a gating structure, derived from the segmented target or surrogate with an applied margin, is tracked, and beam delivery occurs only when the structure is within the defined gating window. However, latency time (the delay between image acquisition, analysis, and machine response) remains a significant concern for gating accuracy [26]. Beyond gating, real-time MLC-based tumor tracking has demonstrated improved target coverage and reduced dose to surrounding normal tissues [27]. Despite its benefits, MRI-guided radiotherapy presents unique challenges, including MRI safety considerations, limited radiation field size, synthetic CT generation for dose calculation, increased system costs, and time-intensive workflows [28,29]. Initial MR-Cobalt systems required up to 90 min per adaptive treatment session, whereas current MR-linac platforms have reduced this duration to a median of 45–53 min [30]. In contrast, recent CBCT-guided adaptive systems like Ethos have reported treatment times under 30 min, improving efficiency without compromising plan quality [31,32].

## 4. Technical Considerations in Adaptive Radiotherapy

One of the most critical technical considerations in adaptive radiotherapy is the quality of online imaging. In MR-guided ART, MRI provides superior soft-tissue contrast without additional radiation exposure and enables continuous intrafraction monitoring. However, a trade-off exists between spatial and temporal resolution, higher image resolution may enhance tracking accuracy but reduces the frame rate [33]. Current research focus on achieving an optimal balance between image speed and quality [34]. In clinical practice, real-time MRI is generally limited to a single 2D imaging plane, which restricts full volumetric motion assessment. Another key trade-off involves magnetic field strength. Higher field strengths improve the signal-to-noise ratio (SNR) due to increased nuclear polarization, but also amplify artifacts such as susceptibility-induced distortions and increase radiofrequency energy deposition in the patient, quantified by the specific absorption rate (SAR) [26,33]. Therefore, quality assurance (QA) to maintain geometric accuracy and ensure MRI safety is essential. In CBCT-guided ART, platforms such as HyperSight CBCT offer rapid image acquisition with image quality approaching that of fan-beam CT. While faster scans can reduce motion artifacts, they may risk incomplete capture of the full motion trajectory [35]. Additionally, current CBCT systems acquire images only at the start of the planning session and immediately before beam delivery, lacking real-time imaging capability to track intrafractional motion or anatomical changes during the adaptive process [24].

Online dose calculation is a critical component of adaptive radiotherapy. In MR-guided ART, magnetic fields alter dose deposition via the Lorentz force, requiring Monte Carlo-based algorithms to model effects like the electron return effect. Efforts to accelerate dose calculation include pre-computed dose kernels [36] and machine learning [37]. MR-only planning, which derives electron density directly from MRI, can eliminate the need for an additional CT simulation. Techniques such as bulk density assignment and deep learning-based synthetic CT generation have demonstrated promising dosimetric accuracy [38,39]. In CBCT-guided ART, deformable registration maps planning CT to daily CBCT for dose calculation, though it may miss changes like air pockets, introducing dosimetric uncertainties [24]. New systems using HyperSight CBCT enable direct, accurate dose calculation on daily images [40]. For both modalities, rigorous QA is essential when using synthetic or deformably registered CT for planning.

While AI and deformable image registration techniques assist in automatic contouring, significant manual adjustments by clinicians are frequently required to ensure accuracy. Contouring remains the most time-consuming and labor-intensive step in the online adaptive workflow, limiting patient throughput and increasing the risk of intrafractional motion during prolonged on-couch time [41,42]. To address this, several strategies have been implemented. In MR-guided systems, only OARs within 2 cm of the PTV are typically contoured and reviewed to reduce workload [43]. CBCT-guided platforms support the use of derived structures pre-defined in the initial plan to facilitate auto-contouring [24]. Ongoing efforts to improve efficiency focus on machine learning-based auto-contouring, making AI-driven segmentation on both MRI and CT a key area of development for streamlining online adaptive workflows [42,44,45].

A key unresolved issue in ART is whether fractional adapted doses can be accurately accumulated voxel-by-voxel using deformable image registration (DIR). This process involves generating displacement vector fields (DVFs) between daily and planning images, deforming the dose using these DVFs, and summing the mapped doses. However, DIR uncertainties remain a major concern. Accurate accumulation requires consistent voxel mapping across images, which is complicated by the inability to uniquely identify voxels within organs and the increasing error associated with large anatomical changes in shape or volume [46]. Validation using DIR quality metrics, such as displacement errors and Jacobian determinants, is essential [47]. Ongoing research focuses on quantifying DIR uncertainty and developing more robust algorithms to improve accuracy and reliability [48,49].

QA is essential for the safe and effective delivery of adaptive radiotherapy (ART). In MR-guided linacs, in addition to routine machine QA, comprehensive MRI-specific QA, similar to that used in diagnostic MRI, is required. Key QA tests include geometric accuracy, MRI image fidelity, and coincidence of MRI and treatment isocenters. All QA devices must be MR-compatible, and certain detectors, such as ionization chambers, may behave differently in magnetic fields, requiring correction factors or modified setups [33,50,51]. Similarly, in CBCT-guided linacs, imaging QA is critical, especially tests for HU constancy, geometric accuracy, and isocenter alignment. For both modalities, end-to-end testing is vital. These tests simulate the entire adaptive workflow using phantoms to detect potential errors in imaging, synthetic CT generation, contouring, dose calculation, and treatment delivery. Deformable phantoms, which mimic anatomical changes, are particularly useful for evaluating auto-contouring, synthetic CT accuracy, and adaptive planning [52,53]. Both systems also require virtual patient-specific QA for adaptive plans, typically involving independent dose calculations to detect gross errors. Meanwhile, measurement-based patient-specific QA is still standard for the initial (pre-treatment) plans [54,55].

## 5. Implementation Challenges

The implementation of ART, including both offline and online approaches, presents significant workflow challenges due to the time-sensitive nature of imaging, contouring, planning, and QA, as well as overall procedural complexity. Offline ART allows for plan adaptation between treatment fractions using conventional workflows and longer turnaround times, making it suitable for addressing gradual anatomical changes such as tumor shrinkage or weight loss. However, it may be inadequate for managing daily or unpredictable variations, and is often hindered by inefficiencies in handoffs, delays in replanning, and the risk of “chasing” anatomy due to lag between simulation and adaptation [56].

In contrast, online ART adapts to daily anatomical changes but imposes high demands on clinical time and staffing. A study evaluating CBCT-guided online ART for anal cancer reported a median (IQR) workflow time of 23.4 (21.4–26.7) min, including imaging, contouring, plan adaptation, and QA, shorter than the approximately 40 min workflow reported in an MRI-guided ART study for rectal cancer [57,58]. This is consistent with our clinical experience with the time required for CBCT-guided OART for anal cancer, with a median (IQR) workflow time of 19.4 (17.8–21.5) min, inclusive of imaging, contouring, plan adaptation and QA [24]. Despite improved efficiency, two recent Failure Mode and Effects Analysis (FMEA) studies identified key implementation risks, including inaccuracies in auto-contouring, challenges in developing robust automated planning templates, and limited integration with existing clinical systems [24,59]. The use of standardized checklists, structured workflows, team training, and targeted QA can substantially reduce risk priority numbers and support the safe integration of online ART.

Although MRI-based ART provides superior soft-tissue contrast, functional imaging, and real-time motion tracking, several key barriers hinder its implementation. The limited speed of volumetric MRI acquisition prolongs treatment sessions. Real-time MRI is limited to two-dimensional acquisitions. System- and patient-dependent geometric distortions due to magnetic field inhomogeneities need to be carefully corrected to maintain spatial accuracy. Integration of MR-derived physiologic imaging and biomarkers, while promising for biologically guided adaptation, also requires standardization and validation. These technological and workflow challenges demand robust QA procedures, multidisciplinary collaboration, and ongoing innovation to enable safe, efficient online ART [60].

Additionally, implementing online ART requires substantial clinical resources, particularly in staffing and training. A nationwide survey of CBCT-guided OART programs found that most institutions treat 1–2 adaptive patients per machine per day, with an average of total session time of 1 h [61]. Radiation therapists typically manage patient setup and imaging, while medical physicists are engaged throughout nearly every stage, including plan verification and treatment adaptation. Physicians play a central role in target delineation and plan approval, but their involvement may vary depending on the institutional protocols. Some centers allow remote review or limit in-person presence during daily sessions. As adaptive treatment volume grows, the need for additional clinical full-time equivalents, particularly among physicians and physicists, increases [62]. A global training commentary further emphasizes shortages and inconsistencies in training for radiation oncologists further challenge widespread online ART adoption and calls for expanded training frameworks to support physician participation within adaptive workflows. These findings highlight the need for staffing strategies and interdisciplinary coordination to enable safe online ART delivery [63].

## 6. Clinical Treatment Planning for Adaptive Radiotherapy in Anal Cancer

Anal cancer is well suited for ART due to the potential for rapid tumor regression during CRT and daily anatomic variations in pelvic OARs, particularly the bowel and bladder. Both systematic and random changes in anatomy can significantly impact dose distribution, underscoring the importance of adaptive techniques to maintain target coverage while minimizing toxicity.

As with standard intensity-modulated radiotherapy, ART begins with a high-resolution planning computed tomography (CT) simulation. Patients are positioned in a reproducible, comfortable setup that can be maintained throughout the treatment course. Target delineation must integrate all available clinical and diagnostic imaging to ensure accurate and comprehensive coverage.

The primary tumor is contoured as the gross tumor volume primary (GTVp), and any grossly involved lymph nodes are defined as GTVn. To account for microscopic disease extension, the GTVp is combined with the entire anal canal and expanded by 1 cm, respecting anatomic barriers such as muscle and bone, to create the primary clinical target volume (CTVp). Similarly, expansions around GTVn generate nodal clinical target volumes (CTVn). An elective CTV is also delineated to encompass regions at risk of subclinical involvement, including the internal and external iliac, perirectal, presacral, and bilateral inguinal nodal basins, extending superiorly to the bifurcation of the iliac vessels at approximately the L5/S1 interspace. Margins of 5 mm are then applied to each CTV to generate planning target volumes (PTVs), which account for uncertainties in patient setup, internal motion, and treatment delivery. Dose prescriptions and fractionation schedules are summarized in Table 2, and coverage goals and OAR dose constraints are listed in Table 3. The goal of treatment planning is to achieve robust PTV coverage while minimizing dose to surrounding OARs such as bowel, bladder, skin, and genitalia.

The choice of adaptation strategy depends on institutional expertise, technology, and available resources. Offline adaptation may be either unplanned or planned. Unplanned offline adaptation is typically triggered by observed systematic changes during the treatment course, such as significant tumor shrinkage, patient weight loss, or shifts in anatomy, that may compromise the original plan’s accuracy. In such cases, a new CT simulation is obtained, and replanning is performed accordingly. Planned offline adaptation can follow predefined strategies, such as a “plan library” approach where multiple plans are generated upfront (e.g., full, partially filled, and empty bladder scenarios), with plan selection tailored to the day’s anatomy as seen on image guidance [64]. Another approach to scheduled adaptation utilizes preplanned imaging during the treatment course, such as mid-treatment PET/CT as in protocols like RTOG 1106 [65]. Online adaptation is typically determined prior to treatment initiation and involves plan modification while the patient is on the treatment table. Once daily setup is complete, a cone beam CT (CBCT) or MRI is acquired. Auto-segmentation tools are used to contour targets and OARs, which are reviewed and edited by a physician to ensure accuracy. A new plan is then generated and compared to the original “scheduled” plan, which is deformably registered to the current anatomy in order to estimate the dose distribution that would result from delivery the original plan unadapted. Currently, two widely adopted autoplanning engines for online ART are Elekta’s mCycle, designed for MR-ART, and Varian’s Intelligent Optimization Engine (IOE), used within the Ethos platform for CBCT-ART. Both algorithms are capable of generating high-quality adaptive treatment plans within minutes while the patient remains on the treatment couch. The mCycle algorithm employs lexicographic multi-criterial optimization (MCO) in combination with Monte Carlo dose calculation, enabling precise dose modulation based on a user-defined hierarchical “wish list” of clinical goals and constraints [66]. In contrast, the Ethos IOE utilizes a prioritized objective-based optimization framework integrated with AcurosXB dose calculation, leveraging disease site-specific clinical templates that encode planning intent and constraint prioritization [67]. Despite differences in optimization methodology and imaging modality, both systems require upfront customization of planning parameters to ensure consistent and clinically acceptable results.

If the adapted plan offers dosimetric advantages, improved target coverage or reduced OAR dose, it is selected for treatment. The plan undergoes an expedited QA process and the treatment is delivered. Images from a representative plan, demonstrating improved target coverage of involved lymph nodes and simultaneous sparing of bowel with an adaptive plan, are shown in Figure 1.

## 7. Clinical Outcomes and Toxicity Benefits

Although clinical outcomes data for adaptive radiotherapy (ART) in anal cancer are still emerging, early evidence suggests that ART offers meaningful dosimetric advantages that may translate into improved toxicity profiles without compromising oncologic efficacy. Most of the current data are derived from feasibility and retrospective studies. For example, Sibolt et al. demonstrated the feasibility of online ART (OART) across various pelvic tumor sites, including anal cancer, with adapted plans selected 100% of the time—primarily to optimize planning target volume (PTV) coverage [68]. Similarly, Åström et al. reported that CBCT-based OART yielded median reductions in bowel bag V4500 cGy by 11.4% and bladder V3500 cGy by 16.1%, primarily through reduced PTV margins enabled by decreased setup uncertainty, without compromising target coverage [57]. Our institutional experience using adaptive radiotherapy for anorectal tumors has also demonstrated significant improvements in PTV coverage for both the primary tumor and elective nodal volumes with adaptive plans compared to scheduled plans [69]. Similarly, review of MR-guided ART for pelvic malignancies shows feasibility of the technique for target delineation and accurate delineation of OARs [70].

While these studies did not assess clinical endpoints, the potential benefit of dosimetric improvements is supported by data from the transition from 3D conformal radiotherapy to intensity-modulated radiotherapy (IMRT). In RTOG 0529, IMRT significantly reduced rates of acute grade ≥ 2 hematologic toxicity, grade ≥ 3 gastrointestinal (GI) toxicity, and grade ≥ 3 dermatologic toxicity, and was associated with shorter treatment interruptions [8]. Subsequent analyses demonstrated that higher bowel V25, V35, V45, and V50 were associated with increased risk of acute grade ≥ 2 GI toxicity [9]. In a prospective study by Han et al., IMRT achieved low rates of acute grade ≥ 3 hematologic, GI, and genitourinary (GU) toxicities, with favorable long-term quality of life outcomes, and identified correlations between volumetric dose constraints and acute toxicities [71].

Currently, two prospective trials—NCT05838391 and NCT05438836—are underway to assess whether the dosimetric benefits of ART will translate into improved clinical outcomes for anal cancer. Additionally, there is growing interest in dose escalation strategies for locally advanced anal cancer, such as in the ACT V trial (ISRCTN88455282). Should this trial demonstrate benefit, ART may prove particularly valuable in enabling safe dose intensification, as has been observed in other disease sites such as pancreatic cancer [72].

## 8. Conclusions

Adaptive radiotherapy is an emerging technology that allows for the rapid adjustment of radiation treatment plans to account for changes in target volumes and normal organ positions. For a radiosensitive disease such as anal cancer, the ability to rapidly re-optimize treatment plans offers the potential to improve target coverage while sparing sensitive normal tissue from radiation doses. Advances in radiation treatment planning and delivery, such as IMRT, have demonstrated significant improvements in treatment toxicity, and ongoing trials are assessing whether the use of cutting-edge adaptive radiotherapy techniques will translate to further improvements in the toxicity profile of this treatment, which is essential to curing anal cancer and the quality of survivorship.

## Figures and Tables

**Figure 1 cancers-17-02478-f001:**
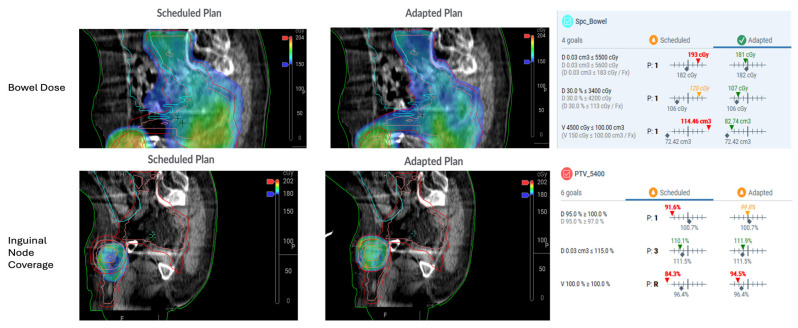
Images from a representative adaptive plan for anal cancer, demonstrating improved coverage of a pathologically enlarged inguinal lymph node, as well as reduced dose to bowel, with adaptive radiotherapy. PTV_5400 = planning target volume receiving 5400 cGy. Spc_Bowel = bowel space. D95% = dose received by 95% of the volume. D0.03 cm^3^ = dose received by 0.03 cm^3^ of the volume. V100% = volume receiving 100% of the radiation dose. D30% = dose received by 30% of the structure. V4500 cGy = volume receiving 4500 cGy or more.

**Table 1 cancers-17-02478-t001:** Adaptive radiation therapy techniques.

Approach	Timing	Use Case	Pros	Cons
Offline ART	Replanning between fractions (e.g., mid-treatment or weekly)	Account for systematic changes over time (tumor shrinkage, weight loss)	Utilizes existing workflow; less resource-intensive than daily adaptation; can selectively adapt for major changes	Cannot respond to random day-to-day variations; optimal timing/thresholds for replanning are unclear
Online ART	Replanning immediately before each fraction (patient on table)	Account for unpredictable interfraction changes (daily organ motion, positioning)	Maximizes precision each fraction; ensures consistent target coverage and OAR sparing despite anatomy changes	Resource and time intensive; requires specialized technology and rapid workflow for imaging, contouring, and QA
Real-Time ART	Adaptation during beam delivery (e.g., gated or beam-tracked radiotherapy)	Account for intrafraction motion (respiration, organ drift during treatment)	Enables margin reduction for moving targets; can “track” tumor motion continuously	Highly technically complex

**Table 2 cancers-17-02478-t002:** Stage-dependent radiation dose recommendations for patients receiving definitive chemoradiotherapy for anal cancer. Adapted from RTOG 0529 and ASTRO clinical practice guideline [2,8].

TNM Stage	Primary Tumor Dose	Involved Node Dose	Elective Nodal Dose
T1–2, N0 With T < 4 cm	5040 cGy in 28 fractions	N/A	4200 cGy in 28 fractions
T2 ≥ 4 cm T3–4, N0	5400 cGy in 30 fractions	N/A	4500 cGy in 30 fractions
Any T with N+ (<3 cm)	5400 cGy in 30 fractions	5040 cGy in 30 fractions	4500 cGy in 30 fractions
Any T with N+ (>3 cm)	5400 cGy in 30 fractions	5400 cGy in 30 fractions	4500 cGy in 30 fractions

**Table 3 cancers-17-02478-t003:** Organ at risk (OAR) dose constraints for patients receiving definitive chemoradiotherapy for anal cancer. Adapted from ASTRO clinical practice guideline [2].

OAR	Dose Constraints
Bowel	D0.03 cc < 5000–5600 cGy
	V4500 cGy < 20–60 cc
	V3500 cGy < 40–150 cc
Bladder	D50% < 3300–4500 cGy
	D5% < 5000–5600 cGy
Femoral head	D50% < 3000–4500 cGy
	D5% < 4400–5500 cGy
Genitalia	D50% < 2000–3500 cGy
Bone Marrow	Mean < 2000–3000 cGy
	V1000 cGy < 70–90%
Skin	D50% < 2000–3500 cGy

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
