# Peer review of "Optimizing Treatment Precision: Role of Adaptive Radiotherapy in Modern Anal Cancer Management"

_cancers, 2025, doi:10.3390/cancers17152478_

Round 1
Reviewer 1 Report
Comments and Suggestions for Authors
Congratulations to the Authors on this interesting review, which addresses a highly relevant topic in clinical practice. Late toxicity resulting from the inclusion of the entire anal canal, and often the vagina, in higher dose volumes during anal cancer chemoradiation has a significant impact on quality of life. The high rate of these adverse effects in long-term survivors is currently prompting us to pay more attention to the management of these patients and to optimising treatment.
Using advanced techniques to reduce the dose to organs at risk (OARs) to improve late toxicity remains a challenge in current radiotherapy, particularly in more advanced cases where dose escalation is needed.
The manuscript is well written, organised, concise and presents excellent work and interesting observations. The tables are clear.
I would suggest:
- To consider including a different but effective adaptive and dose escalation technique, such as sequential brachytherapy boost, in the 'Introduction' paragraph, citing Manfrida S, Fionda B, Mariani S, Luca V, Bertolini R, Barbaro B, Chiloiro G, Frascino V, Tagliaferri L, Gambacorta MA. High-tailored Anal Canal Radiotherapy (HIT-ART): Outcomes of a 10-Year Single Center Clinical Experience. In Vivo. 2024 May-Jun;38(3):1306-1315. doi: 10.21873/invivo.13570. PMID: 38688632; PMCID: PMC11059920 ;
- to Improve the quality of the figure 1;
- to supplement the references in order to emphasise the significant rate of late toxicity and its effect on patients' quality of life (row 39) with: Kronborg CJS, Christensen P, Pedersen BG, Spindler KG. Anorectal function and radiation dose to pelvic floor muscles after primary treatment for anal cancer. Radiother Oncol. 2021 Apr;157:141-146. doi: 10.1016/j.radonc.2021.01.027. Epub 2021 Feb 3. PMID: 33545256; Bentzen AG, Balteskard L, Wanderås EH, Frykholm G, Wilsgaard T, Dahl O, Guren MG. Impaired health-related quality of life after chemoradiotherapy for anal cancer: late effects in a national cohort of 128 survivors. Acta Oncol. 2013 May;52(4):736-44. doi: 10.3109/0284186X.2013.770599. Epub 2013 Feb 26. PMID: 23438358 ; Bentzen AG, Guren MG, Vonen B, Wanderås EH, Frykholm G, Wilsgaard T, Dahl O, Balteskard L. Faecal incontinence after chemoradiotherapy in anal cancer survivors: long-term results of a national cohort. Radiother Oncol. 2013 Jul;108(1):55-60. doi: 10.1016/j.radonc.2013.05.037. Epub 2013 Jul 25. PMID: 23891095;
- to supplement the references on the application of the MRI LINAC in adaptive radiation therapy for pelvic neoplasms. (row 147) Chiloiro G, Gani C, Boldrini L. Rectal Cancer MRI Guided Radiotherapy: A Practical Review for the Physician. Semin Radiat Oncol. 2024 Jan;34(1):64-68. doi: 10.1016/j.semradonc.2023.10.004. PMID: 38105095.
Author Response
Comments 1:
- To consider including a different but effective adaptive and dose escalation technique, such as sequential brachytherapy boost, in the 'Introduction' paragraph, citing Manfrida S, Fionda B, Mariani S, Luca V, Bertolini R, Barbaro B, Chiloiro G, Frascino V, Tagliaferri L, Gambacorta MA. High-tailored Anal Canal Radiotherapy (HIT-ART): Outcomes of a 10-Year Single Center Clinical Experience. In Vivo. 2024 May-Jun;38(3):1306-1315. doi: 10.21873/invivo.13570. PMID: 38688632; PMCID: PMC11059920 ;
Response 1:
Thank you very much for the review and thoughtful comments. We agree with the recommendation. We have added the reference, along with the following sentence, "Additional methods for focal dose escalation, such as brachytherapy boost, have shown promising effectiveness but are available at limited centers."
Comments 2:
- to Improve the quality of the figure 1
Response 2:
Thank you for the feedback. We have added a higher resolution version of the image.
Comments 3:
- to supplement the references in order to emphasise the significant rate of late toxicity and its effect on patients' quality of life (row 39) with: Kronborg CJS, Christensen P, Pedersen BG, Spindler KG. Anorectal function and radiation dose to pelvic floor muscles after primary treatment for anal cancer. Radiother Oncol. 2021 Apr;157:141-146. doi: 10.1016/j.radonc.2021.01.027. Epub 2021 Feb 3. PMID: 33545256; Bentzen AG, Balteskard L, Wanderås EH, Frykholm G, Wilsgaard T, Dahl O, Guren MG. Impaired health-related quality of life after chemoradiotherapy for anal cancer: late effects in a national cohort of 128 survivors. Acta Oncol. 2013 May;52(4):736-44. doi: 10.3109/0284186X.2013.770599. Epub 2013 Feb 26. PMID: 23438358 ; Bentzen AG, Guren MG, Vonen B, Wanderås EH, Frykholm G, Wilsgaard T, Dahl O, Balteskard L. Faecal incontinence after chemoradiotherapy in anal cancer survivors: long-term results of a national cohort. Radiother Oncol. 2013 Jul;108(1):55-60. doi: 10.1016/j.radonc.2013.05.037. Epub 2013 Jul 25. PMID: 23891095;
Response 3:
Thank you for the comment. We agree, and have added the references along with the following sentence, "Impaired anorectal function is impacted by radiation dose and can significantly impact quality of life for patients treated for anal cancer."
Comments 4:
- to supplement the references on the application of the MRI LINAC in adaptive radiation therapy for pelvic neoplasms. (row 147) Chiloiro G, Gani C, Boldrini L. Rectal Cancer MRI Guided Radiotherapy: A Practical Review for the Physician. Semin Radiat Oncol. 2024 Jan;34(1):64-68. doi: 10.1016/j.semradonc.2023.10.004. PMID: 38105095.
Response 4:
Thank you for the comment. We agree, and have added the reference along with the following sentence, "Similarly, review of MR-guided ART for pelvic malignancies shows feasibility of the technique for target delineation and accurate delineation of OARs."
Reviewer 2 Report
Comments and Suggestions for Authors
The review covers the potential role of ART in treatment of anal cancer. The authors deliever a very thorough general-purpose overview of the role and possibilities of ART in radiotherapy. The paper is well written - the part that is specific about anal cancer is due to the scarcity of data rather short compared to the lengthy general description of ART.
I think the paper is publishable as it is.
Author Response
Comments 1:
The review covers the potential role of ART in treatment of anal cancer. The authors deliever a very thorough general-purpose overview of the role and possibilities of ART in radiotherapy. The paper is well written - the part that is specific about anal cancer is due to the scarcity of data rather short compared to the lengthy general description of ART.
I think the paper is publishable as it is.
Response 1:
Thank you very much for the thoughtful review and comments. We appreciate the positive feedback.